# Long-Term Results with Everolimus in Advanced Hormone Receptor Positive Breast Cancer in a Multicenter National Real-World Observational Study

**DOI:** 10.3390/cancers15041191

**Published:** 2023-02-13

**Authors:** Hélène François-Martin, Audrey Lardy-Cléaud, Barbara Pistilli, Christelle Levy, Véronique Diéras, Jean-Sébastien Frenel, Séverine Guiu, Marie-Ange Mouret-Reynier, Audrey Mailliez, Jean-Christophe Eymard, Thierry Petit, Mony Ung, Isabelle Desmoulins, Paule Augereau, Thomas Bachelot, Lionel Uwer, Marc Debled, Jean-Marc Ferrero, Florian Clatot, Anthony Goncalves, Michael Chevrot, Sylvie Chabaud, Paul Cottu

**Affiliations:** 1Department of Medical Oncology, Institut Curie, 26 rue d’Ulm, 75005 Paris, France; 2Centre Léon Bérard, Department of Biostatistics, 69008 Lyon, France; 3Gustave Roussy Cancer Campus, Department of Medical Oncology, 94805 Villejuif, France; 4Centre François Baclesse, Department of Medical Oncology, 14000 Caen, France; 5Centre Eugène Marquis, Department of Medical Oncology, 35000 Rennes, France; 6Department of Medical Oncology, Institut de Cancérologie de l’Ouest, 44800 Saint-Herblain, France; 7Department of Medical Oncology, Institut de Cancérologie de la Méditerranée, 34090 Montpellier, France; 8Centre Jean Perrin, Department of Medical Oncology, 63011 Clermont Ferrand, France; 9Centre Oscar Lambret, Department of Medical Oncology, 59000 Lille, France; 10Department of Medical Oncology, Institut Godinot, 51100 Reims, France; 11Centre Paul Strauss, Department of Medical Oncology, 67200 Strasbourg, France; 12Department of Medical Oncology, Institut Claudius Regaud, 31100 Toulouse, France; 13Centre Georges Francois Leclerc, Department of Medical Oncology, 21000 Dijon, France; 14Department of Medical Oncology, Institut de Cancérologie de l’Ouest, 49055 Angers, France; 15Centre Léon Bérard, Department of Medical Oncology, 69008 Lyon, France; 16Department of Medical Oncology, Institut de Cancérologie de Lorraine, 54519 Nancy, France; 17Department of Medical Oncology, Institut Bergonié, 33076 Bordeaux, France; 18Centre Antoine Lacassagne, Department of Medical Oncology, 06100 Nice, France; 19Centre Henri Becquerel, Department of Medical Oncology, 76038 Rouen, France; 20Department of Medical Oncology, Institut Paoli Calmettes, 13009 Marseille, France; 21Unicancer, Direction des Data, 75013 Paris, France

**Keywords:** advanced luminal breast cancer, everolimus, real-world data, overall survival

## Abstract

**Simple Summary:**

Everolimus is an oral drug used in patients with advanced hormone receptor positive, HER2 negative breast cancer. In this study based on a national French real-world cohort of more than 22,000 patients, we sought to evaluate the impact of everolimus on overall survival. Using statistical methods fit for real-world data, our findings suggest that the use of everolimus may favorably impact overall survival, and that it is very likely underused in this common clinical setting.

**Abstract:**

Everolimus is the first oral targeted therapy widely used in advanced HR+/HER2− breast cancer. We sought to evaluate the impact of everolimus-based therapy on overall survival in the ESME-MBC database, a national metastatic breast cancer cohort that collects retrospective data using clinical trial-like methodology including quality assessments. We compared 1693 patients having received everolimus to 5928 patients not exposed to everolimus in the same period. Overall survival was evaluated according to treatment line, and a propensity score with the inverse probability of treatment weighting method was built to adjust for differences between groups. Crude and landmark overall survival analyses were all compatible with a benefit from everolimus-based therapy. Adjusted hazard ratios for overall survival were 0.34 (95% CI: 0.16–0.72, *p* = 0.0054), 0.34 (95% CI: 0.22–0.52, *p* < 0.0001), and 0.23 (95% CI: 0.14–0.36, *p* < 0.0001) for patients treated with everolimus in line 1, 2, and 3 and beyond, respectively. No clinically relevant benefit on progression-free survival was observed. Causes for everolimus discontinuation were progressive disease (56.2%), adverse events (27.7%), and other miscellaneous reasons. Despite the limitations inherent to such retrospective studies, these results suggest that adding everolimus-based therapy to the therapeutic sequences in patients with advanced HR+/HER2− breast cancer may favorably affect overall survival.

## 1. Introduction

Breast cancer is the second most common cancer worldwide and the most frequent cancer in women [1]. About 70% of breast cancers are hormone receptor positive (HR+) and HER2 negative (HER2−). In patients with advanced HR+/HER2− breast cancer, past and current guidelines strongly recommend endocrine-based treatments unless there is a “visceral crisis”. In addition, European (ABC5) and American (NCCN) recommendations advise exhausting endocrine therapy lines before chemotherapy, again except in cases of rapid progression or endocrine resistance as defined by disease progression in the first 6 months of endocrine therapy for advanced disease [2,3]. However, all patients eventually suffer from progressive disease, and in order to circumvent endocrine resistance, many targeted therapies have been developed. Everolimus, an mTOR inhibitor, was the first targeted therapy to obtain its Marketing Authorization in France (July 2012), for the treatment of patients with advanced HR+ breast cancers resistant to nonsteroidal aromatase inhibitors based on the pivotal Bolero-2 trial [4], and the drug was reimbursed in October 2014. The clinical results were confirmed by many real-world cohort studies across many countries [5,6,7,8,9], all showing a median progression-free survival of 8–9 months in patients with endocrine-resistant metastatic breast cancer (mBC). Most interestingly, the benefit in progression-free survival with everolimus-based therapy appears highly conserved across treatment lines, underlining the consistent efficacy of everolimus in patients with endocrine resistant mBC, and thus suggesting a potential favorable effect on overall survival, however not demonstrated in the Bolero-2 trial.

In 2014, the UNICANCER group (including the 18 French Comprehensive Cancer Centers, which care for over one third of all patients with breast cancer nationwide) launched the Epidemiological Strategy and Medical Economics (ESME) academic initiative in order to investigate real-world data in oncology [10]. Real-world data give the opportunity to retrospectively assess the activity of specific drugs outside clinical trials [11]. Based on this large real-life cohort, the first global results of endocrine therapy sequences have been reported, demonstrating the absence of improvement in overall survival in advanced HR+/HER2− breast cancer [12] while underlining the underuse of endocrine-based therapies in this common clinical setting [13]. These first analyses of the ESME cohort, however, did not specifically describe the evolution of the patients who received everolimus, nor specifically explored the impact on overall survival of specific therapies. We focused the present research on patients treated with an everolimus-based combination. The incidence and context of use of everolimus were presented in an early report [14], suggesting that everolimus was used at some point in less than 20% of patients with HR+/HER2− mBC, mostly with fewer visceral metastasis, mainly in advanced treatment lines, and almost exclusively after official approval by French regulatory authorities. We report here updated data with long-term overall survival analyses, focusing on patients treated from 2012 onwards. The objectives of this study were to describe the impact of everolimus on overall survival and progression-free survival according to treatment line, and to evaluate its positioning in the therapeutic strategy in a real-life setting.

## 2. Materials and Methods

### 2.1. Study Design and Data Source

We conducted a noninterventional, retrospective study to describe the outcome of patients with HR+/HER2− MBC treated with everolimus, selected in the ESME-MBC database. The ESME-MBC database is a national metastatic breast cancer cohort that collects retrospective data using clinical trial-like methodology, including quality assessments. The ESME-MBC database was built from existing information systems, treatment databases, and patients’ electronic medical records, with homogenous onsite-collected information and high-level quality control. The whole methodology was previously extensively detailed in [10].

The present analyses were approved by the Institutional Review Boards of participating institutions. Per French regulations, no formal dedicated informed consent was required, but all patients had approved the use of their electronically recorded data. The ESME analyses were approved by an independent Ethics Committee (Comité De Protection Des Personnes Sud-Est II-2015-79). In compliance with French regulations, the ESME-MBC database was authorized by the French data protection authority and managed by R&D UNICANCER in accordance with the current best practice guidelines [10,12].

### 2.2. Study Population

All consecutive women and men over 18 years diagnosed with HR+ HER2− metastatic breast cancer between January 2012 and December 2017 in the 18 French comprehensive cancer centers were selected (*n* = 7825). Among them, 1897 received at least one dose of everolimus (everolimus) at some point in their therapeutic sequence (study population), 1693 patients were evaluable for successive lines of treatment, and 5928 patients never received everolimus during the course of metastatic disease (comparative population).

### 2.3. Evaluation Criteria

The primary endpoint was overall survival (OS) in patients who received everolimus. Secondary endpoints were the impact of everolimus on overall survival and progression-free survival (PFS) in relation to treatment line (line 1 (L1), line 2 (L2), or line 3 and more (L3+)), the description of patient characteristics at metastatic diagnosis and at the initiation of each treatment line, the position of everolimus in the therapeutic strategy (L1, L2, L3+) and in relation to CDK4/6 inhibitors after 2016 (date of marketing authorization in France), and the description and quantification of the causes of treatment discontinuation.

### 2.4. Statistical Considerations

Descriptive statistics were used to summarize patient characteristics at diagnosis of metastatic disease, and at time of the start of metastatic treatment line (L1, L2, and L3+). Comparisons between everolimus or noneverolimus groups were performed using a chi-square or Fisher’s exact test for categorical data and *t*-test or nonparametric Wilcoxon test for continuous data; a *p* value < 0.05 was considered statistically significant.

Overall survival was defined as the time between the diagnosis of metastatic disease and the date of death (from any cause) or censored to the date of latest news. Progression-free survival was defined as the time from the starting date of treatment until the disease progression or death or the date of latest news. Progression was defined as any of the following events: local/locoregional relapse, progression of known metastases, new metastatic sites, death. A line was defined as a treatment change at least one month after initiation and/or after disease progression. Due to the definition of treatment lines by the ESME team, some patients were not classifiable in lines. Therefore, a treatment initiation more than 12 months after progression was not considered, which explains the final number of patients of 1693.

Both OS and PFS were estimated using the Kaplan–Meier method. The reverse Kaplan–Meier method was used to estimate the median follow-up durations. Hazard ratios are presented with a 95% CI. The landmark approach was used to limit the immortality bias for the analyses of overall survival. We built a propensity score to adjust for differences between groups for specific analyses of OS and PFS according to treatment line, as baseline characteristics of both populations at the initiation of each line could differ according to the chosen treatment. This method reduces biases in the estimation of treatment effects associated with nonrandom observational data (prespecified prognostic factors) and is useful for observational studies in which baseline characteristics differ and when the number of characteristics or potential confounders is relatively large [15,16]. We used the inverse probability of treatment weighting (IPTW) method with stabilized weights on OS [17]. The selected variables in relation to survival outcomes and allocation of everolimus treatment were gender time interval between primary diagnosis and metastatic relapse (de novo metastatic versus <2 years versus >2 years), recurrence (no recurrence versus local recurrence versus loco-regional recurrence), the modality of diagnosis of metastatic disease (systematic examination symptoms), SBR grade (I versus II versus III versus undetermined/not available), age at the initiation of the studied line (<52 years versus ≥52 years), number of metastatic sites at the initiation of the studied line (<3 versus ≥3), and type of metastatic sites at the initiation of the studied line (brain visceral versus non brain visceral versus nonvisceral). Specifically, the logistic model with all covariables gives a propensity score π. For a patient with treatment (everolimus) the weight is $\frac{1}{\pi }$, and for a patient without treatment the weight is $\frac{1}{1-\pi }$. In order to preserve the sample size of the original data, we stabilized weights by using a logistic model without covariable [17]. We ultimately obtained ᴘ, which is the probability of treatment by not taking account of a given covariable. Finally, for a patient with treatment (everolimus) the stabilized weight is $\frac{p}{\pi }$, while for a patient without treatment (everolimus) the stabilized weight is $\frac{1-p}{1-\pi }$. To evaluate the model, we used the Harrell’s C index and a graphic representation of overlapping scores by plotting the kernel density estimate (KDE) of the residuals corresponding to the regression of each component of x on *β*^⊺^ x grouping by the response y. Models with good fit result in plots in which the KDE curves for different values of y are similar in shape and location (see Appendix A). As the log-rank test is inadequate when propensity score weight is taken into account, a robust variance estimator in the Cox model was used.

## 3. Results

### 3.1. Patient Characteristics

Of the 23,698 patients in the ESME-MBC database, 7825 with positive hormone receptor and negative HER2 were diagnosed after 2012, including 1897 who received at least one dose of everolimus. Of these, 1693 patients had identifiable successive lines of treatment (Figure 1, and Table 1, Table 2, Table 3 and Table 4).

Median age at metastatic diagnosis was 63 years (22–03). Patients having received everolimus were slightly younger than noneverolimus patients (25.9% and 23.7% under 52 years, respectively, *p* = 0.057). Everolimus-treated patients had more frequent nonvisceral metastases (52.3% vs. 47.9%, *p* < 0.0001) and bone-only metastases (38.5% vs. 31.3%, *p* < 0.0001) at metastatic diagnosis, and less frequent clinical symptoms (42.4% vs. 47.5%, *p* = 0.0001) than noneverolimus patients. The everolimus–exemestane regimen was the most frequently used endocrine therapy combination (94.3%). Among the 7825 patients, 1897 (24.2%) received at least one dose of everolimus. In the first line setting population, 4.2% received everolimus, 17.9% in the L2 setting population, and 21.4% in the L3+ setting population. Patients who received everolimus as a first treatment line were slightly older (83.6% over 52 years in the everolimus group vs. 76.6% in the noneverolimus group, *p* = 0.01), as well as when everolimus was prescribed in L2 (81.2% over 52 years vs. 76.5%, *p* = 0.005). The age differences between groups was not statistically significant for the L3+ population. Among patients who received everolimus, when prescribed in an L1 setting, 51.6% had nonvisceral metastasis, 35.1% in an L2 setting, and 26.6% in an L3+ setting. Bone-only metastasis at the initiation of everolimus was observed in 39.2%, 24.5%, and 16.9% of patients in L1, L2, and L3+, respectively. Overall, everolimus patients had less frequent additional nonvisceral metastases compared to noneverolimus patients (52.3% nonvisceral metastasis vs. 47.9%, respectively, *p* < 0.0001). In the L3+ population, everolimus patients had less frequent additional nonvisceral metastases compared to noneverolimus patients (26.6% nonvisceral metastasis vs. 17.1%, respectively, *p* < 0.0001) and more frequent bone-only metastasis (16.9% vs. 9.3 respectively, *p* < 0.0001)

### 3.2. Overall Survival

Median follow-up was 47.9 months (0–98.7) and 61.4 months (2.1–98.7) for the overall and everolimus-treated populations, respectively. Median OS in the overall population was 46.8 months (95% CI, 45.5–47.9). Crude and landmark (6 and 12 months) OS analyses all suggested a benefit from everolimus-based therapy (all *p* values < 0.0001). For the everolimus population, the crude HR for overall survival was 0.68 (95% CI: 0.63–0.72) when compared to patients who had not been exposed to everolimus (Figure 2).

The 6-month and 12-landmark OS analyses are shown in Figure 3. For patients with at least a 6-month or 12-month follow-up, 6-month and 12-month HR were 0.74 (95% CI: 0.69–0.80, Figure 3A) and 0.81 (95% CI: 0.75–0.88, Figure 3B), respectively.

To account for imbalance between the everolimus- and non-everolimus-treated groups, we then focused our investigation on adjusted survival analyses, including lines of treatment as a key parameter. Overall, comparing everolimus-treated and non-everolimus-treated patients suggested a striking benefit on overall survival of everolimus-based therapy. Survival curves are presented in Figure 4.

Survival analyses based on IPTW demonstrated a consistent benefit on OS with everolimus treatment when administered in lines 1, 2, and 3+ with respective HR values of 0.34 (95% CI: 0.16–0.72, *p* = 0.0054), 0.34 (95% CI: 0.22–0.52, *p* < 0.0001), and 0.23 (95% CI: 0.14–0.36, *p* < 0.0001). Of note, Harrell’s C index always overlapped for the three analyses, allowing these adjusted analyses (Appendix A).

### 3.3. Progression-Free Survival

Evaluation of PFS according to everolimus-based therapy and line of treatment was an important secondary objective of the study. IPTW analyses suggested a longer progression-free survival with everolimus when administered in the L3+ setting (HR = 0.82 (95% CI: 0.75–0.90), *p* < 0.0001) (Appendix A). However, adjusted PFS for patients receiving everolimus either in line 1 or line 2 were not statistically significant (HR 0.99 (95% CI, 0.84–1.17), *p* = 0.92, and 1.02 (95% CI, 0.94–1.10), *p* = 0.69 respectively).

### 3.4. Treatment Landscape

We examined the landscape of treatment with everolimus in this unselected and large population of patients with advanced ER+/HER2− breast cancer. Causes of everolimus discontinuation were recorded in the 1897 patients having received at least one dose of everolimus. Expectedly, disease progression (54%) and adverse events (26.6%) were the two main causes of discontinuation of everolimus. Other reasons were physician’s choice (7%), patient’s choice (2.3%), and miscellaneous (6.1%). Of note, the median duration of everolimus treatment was remarkably stable across lines of treatment: 5.2 months (interquartile range, IQR, 2.4–10.8), 4.8 months (IQR 2.7–8.9), and 4.8 months (IQR 2.8–8.8) for L1, L2, and L3+, respectively.

We finally focused on the variation of everolimus prescription over time. The prescription rate of everolimus increased from 1.2% in 2012 (when access to the drug was made possible) to 19.5% in 2017 (Appendix A). We observed a limited but steady increase in the proportion of patients receiving everolimus from 2012 to 2017, when CDK4/6 inhibitors became available and were entered into guidelines [3,18]. An exploratory analysis showed that 998 everolimus-treated patients (52.6%) also received a CDK4/6-inhibitor-based treatment, mostly after everolimus therapy (*n* = 826, 87%). Very interestingly, the median duration of CDK4/6 inhibitor therapy for those patients was 4.6 months (IQR 2.9–8.7).

## 4. Discussion

In this study, we harnessed the real-life data from the national ESME program in order to describe the survival outcomes of patients with HR+/HER2− mBC and treated with an everolimus-based combination. We compared the outcomes of these patients to a contemporary population of patients with advanced HR+/HER2− breast cancer, also included in the ESME database but not exposed to everolimus. A striking benefit in overall survival was observed for patients exposed to everolimus, particularly when treated in the second line or third line and beyond settings. In order to limit biases due to the numerical imbalance in some important prognostic parameters such as the metastatic profile at initiation of a line of treatment (Table 2, Table 3 and Table 4), we developed adjusted survival analyses based on a propensity score and adjustment based on inverse probability of treatment weighting. These techniques are widely used and recognized as powerful tools for the analysis of real-world cohorts [19,20], and the present report is the first to use such methods to evaluate the clinical utility of everolimus in a real-world setting, together with a very-long-term follow-up. We included many potential confounding factors such as gender, time interval between primary diagnosis and metastatic relapse, recurrence, metastatic disease diagnosis context, SBR grade, and age, number, and type of metastatic sites at the initiation of the line of interest. Very interestingly, IPTW-adjusted analyses for OS confirmed a benefit from the everolimus-based therapy when administered in lines 2 and 3. In the first line setting, the HR was 0.34 (95% CI, 0.16–0.72) with a *p* value of 0.054, a trend for benefit of borderline significance.

Furthermore, IPTW analyses suggested a significantly longer progression-free survival with everolimus when administered in an L3+ setting. More globally, it is striking to observe that the initial [13] and presently updated ESME real-word data are in line with the survival outcomes that were reported in the prospective trials, whether everolimus was combined with exemestane [4], tamoxifen [21], or fulvestrant [22]. The BOLERO-2 final PFS analysis showed a median PFS of 7.8 months at median follow up of 18 months in a nonsteroidal aromatase-inhibitor-resistant population with the everolimus–exemestane combination. As for the GINECO [21] and TREND [22] studies, the median PFS were 8.6 months and 7.4 months, respectively, in postmenopausal women with hormone receptor-positive, HER2-negative, aromatase-inhibitor-resistant mBC. Our data also confirm the clinically meaningful benefit from the everolimus-based therapy found in the BALLET study [6], among patients with advanced HR+/HER2− mBC.

We also looked at to how everolimus was used in the successive therapeutic sequences, including during the early phase of the CDK4/6 inhibitors era. It is somewhat startling to observe that, at the time when everolimus was the first and only approved targeted therapy in advanced HR+/HER2− breast cancer, it was prescribed at some point in the course of the disease in only 24.2% of all cases, and mainly in the L3+ setting. In line with the consistent PFS and OS results, the median duration of treatment with everolimus was very stable at about 5 months for each treatment line, including very-long-term responders in very advanced patients. Remarkably, all these results complement an earlier ESME report [12] that demonstrated that overall survival did not improve in the decade preceding the introduction of CDK4/6 inhibitors. Taken together, these data strongly suggest that everolimus is a valuable and underused drug in patients with advanced HR+/HER2− breast cancer. Lastly, it seems interesting to note that in patients who received both CDK4/6 inhibitors and an everolimus combination (mostly CDK4/6 inhibitors after everolimus at the time of the study), the median duration of therapy with CDK4/6 inhibitors was much shorter (4.8 months) than observed in the literature [23,24,25,26]. This might again strengthen the potential interest in positioning everolimus later in the disease course, as also suggested by recent reports [7,9,27]. This is in accordance with recommendations suggesting a prescription of everolimus from the second line and later during the course of the disease [18].

We are fully aware of the limitations of the present report. Our study is limited by its retrospective nature, and the lack of individual information on important clinical features. For instance, patient weight variations, performance status, or LDH levels at the time of metastatic disease diagnosis or at each treatment line appeared to be scarcely collected in electronic medical records and are consequently not exploitable. Furthermore, the potential toxicity of everolimus needs to be taken into account, as in this series the treatment was discontinued in 26.6% of cases owing to adverse events, although it might be mitigated by individual dose escalation [28]. Consequently, patient selection by oncologists, according to individual risk–benefit balance, may favor the fittest, which could have an influence on overall survival. Likewise, despite adjusted IPTW analyses, our findings seem to suggest that the patients who received everolimus had less severe diseases, which might also explain part of the benefit observed on overall survival. Nevertheless, the adjusted PFS analyses showed this same benefit, suggesting an obvious contribution of everolimus to survival outcomes. Finally, the clinical utility of everolimus in patients pretreated with CDK4/6 inhibitors remains to be thoroughly evaluated [29].

## 5. Conclusions

Taken together and despite the limitations of such retrospective real-world data, it is our belief that the present report brings important information on the role of everolimus in the management of patients with advanced HR+/HER2− breast cancer. The favorable outcomes we report here in a large real-life population suggest a strong benefit from everolimus-based therapy. These data support current guidelines and should prompt oncologists to consider including everolimus in the therapeutic strategy for patients with advanced HR+, HER2− breast cancer, particularly from the third line onwards, while the benefit–risk balance must be assessed on a case-by-case basis.

## Figures and Tables

**Figure 1 cancers-15-01191-f001:**
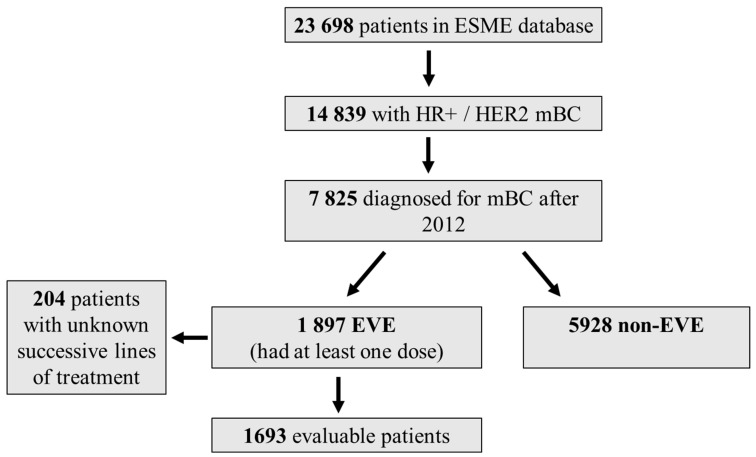
Patient disposition. Patient characteristics at metastatic diagnosis and at the start of each therapeutic line are detailed in Table 1 and in Table 2, Table 3 and Table 4.

**Figure 2 cancers-15-01191-f002:**
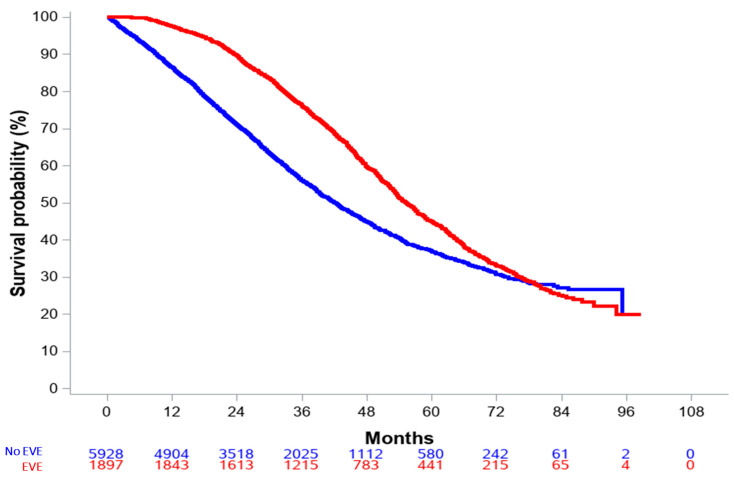
Overall survival in the entire population, according to everolimus exposure.

**Figure 3 cancers-15-01191-f003:**
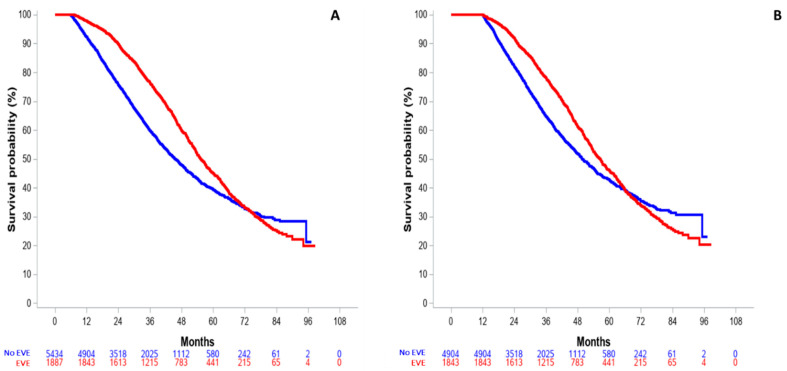
Landmark analysis of overall survival in the entire population, according to minimal follow-up. (**A**) 6-month landmark OS. (**B**) 12-month landmark OS.

**Figure 4 cancers-15-01191-f004:**
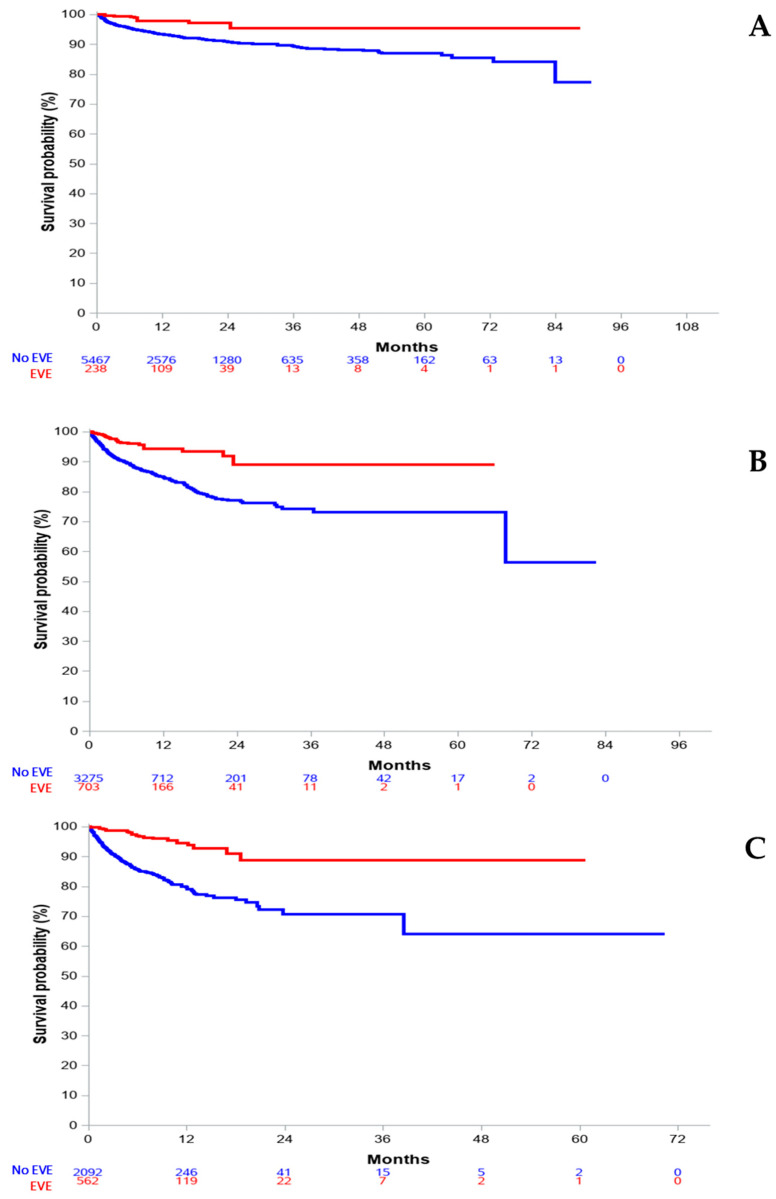
IPTW-adjusted overall survival, according to line of treatment with everolimus. (**A**) first line; (**B**) second line; (**C**) third line and beyond.

**Table 1 cancers-15-01191-t001:** Patient characteristics at onset of metastatic disease, according to everolimus exposure.

Demographic Variables	All (*n* = 7825)	No Everolimus (*n* = 5928)	Received at Least One Dose of Everolimus (*n* = 1897)	*p* Value *Chi2 or Fisher*
Age (years) at metastatic disease diagnosis, *n* (%)				0.0574
<52	1898 (24.3)	1407 (23.7)	491 (25.9)	
≥52	5927 (75.7)	4521 (76.3)	1406 (74.1)	
Median age, years (range)	63 (22–103)	64 (22–103)	61 (26–93)	<0.0001
Number of metastatic sites at metastatic disease diagnosis, median (min–max)	1.00 (1.00–11.00)	1.00 (1.00–11.00)	1.00 (1.00–7.00)	0.5263
Type of metastases, *n* (%)				<0.0001
Brain metastases	268 (3.4)	236 (3.9)	32 (1.7)	
Non brain visceral metastases	3728 (47.6)	2855 (48.2)	873 (46)	
Nonvisceral metastases	3829 (48.9)	2837 (47.9)	992 (52.3)	
Bone-only metastases, *n* (%)				<0.0001
No	5237 (66.9)	4071 (68.7)	1166 (61.5)	
Yes	2588 (33.1)	1857 (31.3)	731 (38.5)	

**Table 2 cancers-15-01191-t002:** Patient characteristics at the initiation of line 1.

Demographic Variables	All (*n* = 6012)	No Everolimus in L1 (*n* = 5762)	Received at Least One Dose of Everolimus in L1 (*n* = 250)	*p* Value *Chi2* or *Fisher* (*F*)
Age (years) at the initiation of L1, *n* (%)				0.0111
0–52	1384 (23)	1343 (23.3)	41 (16.4)	
≥52	4628 (76.9)	4419 (76.7)	209 (83.6)	
Number of metastatic sites at the initiation of L1, median (min–max)	1.00 (1.00–8.00)	1.00 (1.00–8.00)	1.00 (1.00–5.00)	0.0135
Type of metastases, *n* (%)				0.0912
Brain metastases	249 (4.1)	244 (4.2)	5 (2)	
Non brain visceral metastases	2966 (49.3)	2850 (49.5)	116 (46.4)	
Nonvisceral metastases	2797 (46.5)	2668 (46.3)	129 (51.6)	
Bone-only metastases, *n* (%)				0.0026
No	4172 (69.4)	4020 (69.8)	152 (60.8)	
Yes	1840 (30.6)	1742 (30.2)	98 (39.2)	

**Table 3 cancers-15-01191-t003:** Patient characteristics at the initiation of line 2.

Demographic Variables	All (*n* = 4189)	No Everolimus L2 (*n* = 3439)	Received at Least One Dose of Everolimus L2 (*n* = 750)	*p* Value *Chi2* or *Fisher* (*F*)
Age (years) at the initiation of L2, *n* (%)				0.0054
0–52	949 (22.7)	808 (23.5)	141 (18.8)	
≥52	3240 (77.6)	2631 (76.5)	609 (81.2)	
Number of Metastatic Sites at the initiation of L2, median (min–max)	2.00 (1.00–8.00)	2.00 (1.00–8.00)	2.00 (1.00–7.00)	<0.0001
Type of metastasis, *n* (%)				<0.0001
Brain metastasis	327 (7.8)	296 (8.6)	31 (4.1)	
Non brain visceral metastasis	2684 (64.1)	2228 (64.8)	456 (60.8)	
Nonvisceral metastasis	1178 (28.1)	915 (26.6)	263 (35.1)	
Bone-only metastases, *n* (%)				
No	3471 (82.9)	2905 (84.5)	566 (75.5)	
Yes	718 (17.1)	534 (15.5)	184 (24.5)	

**Table 4 cancers-15-01191-t004:** Patient characteristics at the initiation of line 3 and beyond.

Demographic Variables	All (*n* = 2795)	No Everolimus L3 (*n* = 2198)	Received at Least One Dose of Everolimus L3 (*n* = 597)	*p* Value *Chi2* or *Fisher*
Age (years) at the initiation of L3, *n* (%)				0.1770
0–52	667 (23.9)	537 (24.4)	130 (21.8)	
≥52	2128 (76.1)	1661 (75.6)	467 (78.2)	
Number of metastatic sites at the initiation of L3, median (min–max)	3.00 (1.00–8.00)	3.00 (1.00–8.00)	2.00 (1.00–8.00)	<0.0001
Type of metastasis, *n* (%)				<0.0001
Brain metastasis	329 (11.8)	286 (13)	43 (7.2)	
Non brain visceral metastasis	1932 (69.1)	1537 (69.9)	395 (66.2)	
Nonvisceral metastasis	534 (19.1)	375 (17)	159 (26.6)	
Bone-only metastases, *n* (%)				
No	2489 (89.1)	1993 (90.7)	496 (83.1)	
Yes	306 (10.9)	205 (9.3)	101 (16.9)	

## Data Availability

No new data were created or analyzed in this study. Data sharing is not applicable to this article.

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
