# Peer review of "Long-Term Results with Everolimus in Advanced Hormone Receptor Positive Breast Cancer in a Multicenter National Real-World Observational Study"

_cancers, 2023, doi:10.3390/cancers15041191_

Round 1

Reviewer 1 Report

In this retrospective study from a French data set, the authors show that ER+/HER2- metastatic breast cancer patients exposed to everolimus have a better survival than unexposed patients.

The study has many biases as described in the conclusions, but above all it does not allow us to state that the increase in OS is due to the use of everolimus.

In fact, in PFS analyses, everolimus only when used beyond the third line appears to have minimal PFS advantage after IPTW.

Therefore, these results do not allow us to state that the use of everolimus increases the survival of patients with ER+/HER2+ metastatic breast cancer (as not demonstrated by the BOLERO-2 study), but they do allow us to most likely highlight how patients who are exposed to multiple hormonal lines have a better survival outcome.

This could most likely be related to favorable disease characteristics.

In the light of these considerations, in order to consider accepting the job, it would be useful to carry out an analysis of the hormonal lines performed by the patients and evaluate their prognostic impact by analyzing them in multivariates with the impact of everolimus. So, depending on the results obtained, modify the conclusions.

The quality of the images is not adequate and the x and y axis caption is incorrect (to be corrected with x: months, y: survival probability (%)).

Reviewer 2 Report

the limitations have to be rewritten  .

Specifically though the limitations are quiet understandable and reasonable but need to be re worded 

Specifically

line 332 - more description & justification for not having record on performance status

line 341 342 - We also acknowledge that the benefit of everolimus in  patients pretreated with CDK4/6 inhibitors remains to be thoroughly evaluated. - ALSO to be reworded

These two lines though  facts are giving too much negative effect to the entire study hence to be re written.

Reviewer 3 Report

Everolimus is one of the strong immunosuppressors widely used in different areas of medicine including transplantation. Its role in breast cancer treatment is not yet well established, but could be a future attractive opportunity for both HR-positive and HR-negative breast cancer subtypes.

The research of novel therapeutic strategies and tailored treatment in breast cancer is ongoing and is of an utmost importance as breast cancer is known to be a most diffuse female tumor in the world. The retrospective evaluation of everolimus-based therapy effect may open prospectives to an additional window of opportunity in treating HR+/HER2- breast cancer, hence these data is a valuable input. 

Adding prospective studies could be helpful to establish the effect of this therapy, monitoring carefully the patients' status, but enrichment of  retrosperctive part data is a good start.

The article is structured in a concise manner with a good language, updated literature sources and comprehensible results representation, including table and figures, followed by logical well-shaped conclusions.

Thanks to authors.

Reviewer 4 Report

The authors conducted a study for evaluating the impact of everolimus on overall survival. The results they presented seem enough to support their conclusions. However, there are several questions should be addressed:

1.    The paper looks like a clinical report rather than an academic article. What are the unique contributions of this work? The authors claimed that “Using statistical methods fit for real world data, our findings suggest…”, what is the key novelty of this work comparing to related studies. As far as I know, there have been many studies reporting the effect of everolimus on breast cancer.

2.    Abstract: The authors should briefly mention the limitations of related works in Abstract, so that readers can get the key contributions of this work easier.

3.    Introduction: It seems more references should be involved in introduction to help readers understand research background.

4.    Table 4. “Patient characteristics at the initiation of line 3and beyond.” I guess there should be a space between ‘3’ and ‘and’.

5.    Fig. 3: The font of numbers in this figure is too small. It is suggested to make it larger.

6.    Has the research protocol been proved by any ethics committee? It seems the authors didn’t mention it in the manuscript.
